# On the Thermal Behavior during Spatial Anisotropic Femtoseconds Laser-DNA Interaction: The Crucial Role of Hermite Polynomials

**DOI:** 10.3390/ma16093334

**Published:** 2023-04-24

**Authors:** Mihai Oane, Cristian Nicolae Mihailescu, Alexandra Maria Isabel Trefilov

**Affiliations:** National Institute for Laser, Plasma and Radiation Physics, 077125 Măgurele, Romania

**Keywords:** deoxyribonucleic acid, Hermite-Gauss laser modes, fs laser pulses, heat equation, DNA elongation, cancer treatment

## Abstract

A novel analytical formalism based on the quantum heat transport equation is proposed for the interaction of fs-laser pulses with deoxyribonucleic acid (DNA) strands. The formalism has the intensity of the laser beam and the interaction time between the laser and the DNA as input parameters. To this end, the thermal distribution generated in the irradiated DNA strands was introduced by splitting the laser beam into transverse Hermite-Gauss modes. To achieve this goal, a new powerful mathematical model was developed and applied. Fluctuations in laser intensity were taken into account by modeling them as superpositions of Hermite-Gauss laser modes. These analyses were carried out for a laser pulse duration of 100 fs, where a tiny heat-affected zone is expected, with positive predicted effects on the stability and repeatability of this technology. The main conclusion is that the laser beam spatial distribution intensity plays an essential role in the generation of the shape and magnitude of the thermal field at the junction of the irradiated DNA strands. The model may prove useful in modeling laser beam processing under significant intensity fluctuations. There are at least two main areas of application for the present model of heat transfer from laser to DNA: (i) the study of DNA elongation without destroying the target information (for a sample temperature variation lower than 10 K; in the case of H[1,y]); and (ii) cancer treatment (especially of skin tissue), where we should obtain a temperature variation higher than 10 K (but lower than 30 K; in the case of H[2,y], H[4,y]), in order to eradicate the diseased cells.

## 1. Introduction

DNA is an organic polymer composed of a sugar (deoxyribose)-phosphate backbone and four nitrogenous base subunits: guanine (G), cytosine (C), adenine (A), and thymine (T). The subunits, called nucleotide monomers, are connected by strong electromagnetic attractions [1]. The structure of DNA, as depicted in the Watson and Crick model, is a double-stranded, antiparallel, right-handed helix. The sugar-phosphate backbones of the DNA strands form the exterior of the helix, while the nitrogenous bases are on the inside, forming hydrogen-bonded pairs that hold the DNA strands together. [2]. Thus, the DNA double helix is stabilized by two main forces: hydrogen bonds between nucleotides and base nucleotides and base-stacking interactions between aromatic nucleobases [3].

The Watson and Crick model explains the complementarity of the base sequences on the two strands and, implicitly, the heritability of genetic information and the transcription of genetic information, but it does not consider other advantages, such as information accessibility and packaging.

Today, other molecular simulation models of nucleic acids are well established, covering a wide range of information, while modeling techniques are a powerful tool for describing them. All these models are required to follow two rules: (i) the models should be consistent with the physical rules that govern the structure of molecules, and (ii) they should explain all the experimental information that is available [4,5].

DNA models can be better understood by studying molecular electronics, the ‘making’ of electronics at the molecular level, which can be used as a computational tool in its own right. Clearly, molecular electronics belong to an area of science in which mathematical modeling and simulations are powerful tools for classical, semi-classical, and microscopic level calculations. The studies show that the models cleverly mix macroscopic and microscopic notions at the semi-classical level, which is also the focus of this paper. Indeed, there is a junction between classical notions (such as the heat transport equation) and those of quantum mechanics at the level of a few or single molecules [6,7].

In this regard, the study of DNA involves three levels of study: classical (macroscopic), semi-classical, and microscopic (quantum). Classical models assume that DNA is an ideal rod that obeys macroscopic laws. However, these models do not take into account the sequence structure of DNA.

The semi-classical approach assumes that many small rods form a normal rod. These small rods are entitled beads. They interact individually with external forces.

Semi-classical models allow experimental data on polynucleotides to be linked to their macroscopic properties. From a theoretical point of view, we can use semi-classical heat equations with quantum “ingredients”. The model presented in the present study can be regarded as a semi-classical approach to the thermal effects of laser-DNA interaction [8].

At the molecular scale, the heat and charge transport phenomena through molecules can be regarded as quantum phenomena, where the electrons are the charge and heat carriers [9,10]. Many experiments have been carried out in which an electric current is passed between two metallic electrodes via a single-layer molecular film [11,12,13]. The transport phenomena can be examined by scanning tunneling microscopy with a single or a few molecules on a gold surface, with the tip acting as a counter electrode [14,15]. Several studies have been carried out on the current through a single molecule, with the connection to both electrodes placed symmetrically to a well-defined chemical bond, allowing the function to remain mechanically stable even at room temperature [16,17]. However, all these studies must take into account the irreversible chemical damage produced in the DNA’s internal structure due to abnormal factors. One of these factors is temperature; above 190 °C, complete DNA degradation takes place [18].

Lately, ultrafast laser pulse excitation has been used to generate highly localized temperature elevations capable of controlling transport phenomena in single chemical molecules and hydrogen molecules [11,19,20]. It is therefore necessary for the models to be able to image processes at the atomic level. The first measurements of dynamic processes deep inside an atom were carried out using ultrafast spectroscopy [21]. In this context, the paper introduces the thermal evolution in the case of laser-DNA interaction using a new approach based on Hermite polynomials. It is known that there is always a zone surrounding laser irradiation spots, called the heat-affected zone (HAZ), where either metal or thermoplastics, or smaller proportions of other materials, are not melted, even though their microstructure and properties change [22]. The heat from the welding process and subsequent cooling cause this change from the weld interface to the end of the HAZ. The extent and magnitude of the property change depend primarily on the base material and the amount and concentration of heat input. The thermal diffusivity of the base material plays a large part in this change—if the diffusivity and the material cooling rate are high, then the HAZ is relatively small. Alternatively, low diffusivity leads to slower cooling and a larger HAZ. The amount of heat input reflected in the laser pulse duration plays a key role in this process, as the shorter the duration of the laser pulse, the smaller the HAZ extension will be. This makes it of particular interest to carry out laser heating with shorter and shorter laser pulses, down to the 100 fs limit considered in these studies [23].

On the other hand, a limited, quite small HAZ confers high precision to the laser processes and high stability to the laser induced processing/heating. The reported results can be easily reproduced by any laboratory with minimum manufacturing costs. It should also be noted that laser pulses of 100 fs duration are available on the market, although much shorter laser pulses (i.e., as) have been reported. This means that this study is in line with the trend of research and progress in the field.

The model proposed in this paper could be extended to other types of radiation with multiple applications, such as those used to treat cancer. Laser cancer therapy is very popular because of the great advantages of high-energy particle irradiation (with an order of magnitude of MeV), which produces a Bragg peak. As we have seen, up until now there have been major efforts to model and understand the interaction between radiation and biological targets such as cells, tissues, and DNA [24,25,26,27,28,29,30,31]. The first step in understanding the phenomenon is to evaluate the cross section of the irradiation of biomolecules and establish the mechanism of the interaction with secondary electrons induced by the incident radiation.

The models and methods of cross-section evaluation are very few and complex. It is important to remember that in microscopic models, particles obey the rules of quantum mechanics. In view of all the above information, the present proposal requires a good understanding of how the target biomolecules respond to the temperature increase during fs laser pulse hitting, especially for low-energy electrons, and thus a good comprehension of the binding energies. This contribution therefore aims to complement the existing knowledge with a completely new, powerful mathematical model to elucidate and explain the thermal behavior under irradiation. The model describes the interaction of ultra-short (fs) laser beams of asymmetric moderate intensity with the DNA molecules to analyze the thermal changes and proposes a model to eradicate the diseased localized cancer cells that show no signs of having spread. Here, the thermal waves generated in the irradiated DNA strands were analyzed by considering the laser beam in terms of decoupled transverse Hermite-Gauss modes. The model was used to derive a highly accurate formula to describe the temperature field in DNA samples during fs laser irradiation. The model is simplified and does not take into account the internal structure of DNA, as no significant modifications are expected when the results are compared with nucleotide sequences with enriched cytosine-guanine or adenine-thymine partitions. In addition, the model has a fast simulation time of less than 1 min, and can be easily adapted for more complex biological simulations.

## 2. Mathematical Modeling

A schematic of the interaction between a laser and two strands of DNA is shown in Figure 1. Although DNA molecules are symmetrical along the y and z axes, the system behaves asymmetrically because the direction of the laser beam propagation is along the *z*-axis. It is assumed that the laser beam is incident at the center of the two DNA strands, while the laser beam is stationary at the given point.

The application of the quantum heat transport equation [32] is extended to the study of heat generation and transport in DNA structures by ultra-short laser pulses. Specifically, a new algebraic approach has been used regarding the polynomial decomposition of Hermite-Gauss modes:(1)$$T\left(x,y,z,0\right)\propto \underset{m}{\sum }\underset{n}{\sum }F\times f_{1}(x)\times f_{2}(y)\times f_{3}(z)$$
where:(2)$$f_{1}\left(x\right)\equiv {\left[H_{m}\left(\frac{\sqrt{2}x}{w}\right)\right]}^{2}\times exp{\left[-\left(\frac{x^{2}}{w^{2}}\right)\right]}^{2}$$
(3)$$f_{2}\left(y\right)\equiv {\left[H_{n}\left(\frac{\sqrt{2}y}{w}\right)\right]}^{2}\times exp{\left[-\left(\frac{y^{2}}{w^{2}}\right)\right]}^{2}$$
(4)$$f_{3}\left(z\right)\equiv exp\left[-\alpha z\right]$$
where *x*, *y*, *z*, and *t* stand for Cartesian coordinates and, respectively, time; $H_{m}\left(\frac{\sqrt{2}x}{w}\right);H_{n}\left(\frac{\sqrt{2}y}{w}\right):$ are Hermite polynomials of *m* and *n* orders, respectively; *T* represents the temperature increase rather than the absolute one; *F* is the laser fluence and w is the *1*/*e* radius of the laser spot (also known as the laser beam waist, w = $w_{x}=w_{y}$), and α designates the absorption coefficient in the case of the laser-DNA interaction, which is assumed, in a first approximation, to be equal to the laser-water absorption coefficient.

According to the general mathematical theory of Hermite polynomials for m and n integers:(5)$$H_{m}\left(x\right)=a_{m}x^{m}+a_{m-2}x^{m-2}+\cdots +a_{2}x^{2}+a_{0}$$
(6)$$H_{m}^{2}\left(x\right)=a_{m}^{2}x^{2m}+\cdots +a_{0}^{2}$$
(7)$$H_{n}\left(x\right)=a_{n}x^{n}+a_{n-2}x^{n-2}+\cdots +a_{1}x$$
(8)$$H_{n}^{2}\left(x\right)=a_{n}^{2}x^{2n}+\cdots +2a_{1}a_{3}x^{4}+a_{1}^{2}x^{2}$$

In Equations (5)–(8), the coefficients $a_{m},\ldots ,a_{2},a_{0},a_{n},\ldots ,a_{3}\;and\;a_{1}$ are real integers.

It is easy to check that both the Gaussian and the constant (and quasi-constant) parts of the Hermite polynomials have circular symmetry. This follows from Equations (1)–(8), assuming that *x*, *y*, *z*, and w≪1.

For the numerical simulations, we considered that: *y* and *z* are in the range $\left[-1,1\right]\;µm$, and w = 2 μm.

According to the current literature, the diameter of DNA molecules covers a rather large range, from several nm to tens of μm, as it is thought to contain several random coil conformations in solution due to entropic elasticity and thus contain a number of more than 7000 DNA double strands [23]. The model therefore assumes that the length of the DNA molecule is 50 μm.

It is presumed that the center of the laser beam (*x*, *y*, *z*) = (0, 0, 0) interacts with the DNA at the junction of the strands. In this study, we have assumed that the experimental cavity is filled with water and has the following dimensions: *h* = 10 µm, *L* = 100 µm, and the thickness is equal to 2 µm, as shown schematically in Figure 1.

R. Bashir mentioned [32] that DNA does in fact behave like a good heat conductor. It was assumed that two complementary DNA strands could form a molecular switch that operated between high and low resistance states. This is now regarded as a key ‘’device’’ in molecular electronics [33,34,35,36,37].

If the DNA switch is irradiated with ultra-short (i.e., fs or as) laser pulses, the 1-D heat equation, according to references [38,39,40,41], is as follows:(9)$$\frac{\partial^{2}T}{{\partial t}^{2}}+\frac{{v^{2}m}_{e}}{ћ}\frac{\partial T}{\partial t}+\frac{2V{v^{2}m}_{e}T}{{ћ}^{2}}-v^{2}\frac{\partial^{2}T}{{\partial x}^{2}}=0$$
where T is the temperature variation; $m_{e}$ stands for the mass of the heat carriers in DNA, which are actually electrons; X indicates the space coordinate; *V* represents the potential carrier generated at the interface; t is the time, ν is the thermal wave velocity, and ћ is the reduced Max-Planck constant.

Equation (9) derived by Kozlovski et al. [38,39] can be rewritten as:(10)$$\frac{\partial^{2}T}{\partial t^{2}}+\varepsilon \frac{\partial T}{\partial t}+kT-\alpha \frac{\partial^{2}T}{\partial X^{2}}=0$$
where:(11)$$\varepsilon \equiv \frac{{\nu^{2}m}_{e}}{ћ}$$
(12)$$k\equiv \frac{2V{\nu^{2}m}_{e}}{{ћ}^{2}}$$
(13)$$\alpha \equiv \nu^{2}$$

The source term, with respect to the boundary conditions, has been defined according to Equations (1)–(8) and (14):(14)T(x,y,z,∞)=0

According to Zhukovsky et al. [42], the solution of Equation (10) can be derived with the help of the heat operator technique and is as follows:(15)Tx,t=e−εt2×t4π∫0∞dξξ32×e−t216ξ−ξε2+4k×S^fx

Considering the heat operator Ŝ as:(16)$$\hat{S}f\left(x\right)=e^{-4\alpha \xi \partial_{x}^{2}}\times f\left(x\right)$$
where *T* is the temperature variation, t represents the time, ξ indicates the increment term, Ŝ stands for the heat operator, and ε, k, and α are defined by Equations (11)–(13). In these equations, f(*x*) is the source term, which is proportional to the incident power.

The aim was to derive an analytical formula for the most general case of an incident laser beam, given the solution to a term describing the source of the form $x^{m}$. This is due to the linear superposition properties of the heat equation solutions. Considering the equations presented in Ref. [42], the general form (for the source term: $x^{m}$) is given by:(17)$$T\left(x,t\right)=\frac{te^{-\frac{t\varepsilon }{2}+\gamma \times \left(x\right)}}{4\sqrt{\pi }}\times \left[\int_{0}^{\infty }\frac{du}{u^{3/2}}\times e^{-\frac{t^{2}}{16u}-n\delta }\right]{\times H}_{m}\left(x,0\right)$$
where:(18)$$\delta =\varepsilon^{2}+4(K+\alpha \gamma^{2})$$

Here $H_{m}\left(x,0\right)$ is the m-kind Hermite polynomial with two variables, when y = 0. For example (m∈{1,2,3,4,5}):(19)$$H_{0}^{2}\left(x,0\right)=1$$
(20)$$H_{1}^{2}\left(x,0\right)=2x$$
(21)$$H_{2}^{2}(x,0)=4x^{2}-2$$
(22)$$H_{3}^{2}(x,0)=8x^{3}-12x$$
(23)$$H_{4}^{2}(x,0)=16x^{4}-48x^{2}-12$$
(24)$$H_{5}^{2}\left(x,0\right)=32x^{5}-160x^{3}-120x$$

## 3. Simulations

The interaction between the laser beam and two strands of DNA was simulated using the “MATHEMATICA” software 11. The main input data are compiled in Table 1 [39]. It is further assumed that the incident laser power was of *p* = 1 W and the beam waist was w = 50 µm. A Core i7, 8th generation computer with 32 GB of RAM was used for the calculations. Computation times were generally less than 20 s.

It is estimated that the HAZ, in this case, is (χτ)^0.5^. In this simulation, χ = 0.143 m^2^s^−1^ was estimated based on the assumption that it is the same as in the case of water at 25 °C and τ = 10–13 s, due to the fact that more than 73% of the human soft man body tissue is composed of water [40]. The model simulation therefore assumes that the thermal parameters of the tissue are identical to those of water.

This gives the result HAZ = 0.13 μm, which is no more than 2.6 × 10^−3^ (%) with respect to the beam radius of 25 μm. Therefore, our estimates of the temperature distribution under laser irradiation are highly accurate.

It should be noted that for ns laser pulses, the HAZ increases significantly to 75 μm or more. This significantly reduces the time and space accuracy of the simulations.

## 4. Results and Discussion

Figure 2a, Figure 3a and Figure 4a show the functions H[1,y], H[2,y], and H[4,y] respectively. These functions are very important in the final solution of the heat equation. 3D, 2D, and 1D thermal fields versus x and irradiation time, t, are displayed in Figure 2, Figure 3 and Figure 4 (b), (c), and (d), respectively. In fact, the *X*-axis is the only one where the laser beam asymmetry can manifest, while along the *Z*-axis one has constant or close to zero values for the laser intensity variation of the strand junction of DNA. It should also be noted that for a very small y, the Hermite function becomes either very close to zero (for n—odd) or constant (for n—even). Note that T in the model equations is actually dT in Figure 2, Figure 3 and Figure 4. Comparing Figure 2 with Figure 3 and Figure 4, it can be concluded that the thermal fields generated by type H[1,y] (n-odd) are suitable for laser stretching of DNA, and those of H[2,y] and H[4,y] are excellent for cancer treatment. The thermal fields generated in the cases of H[2,y] and H[4,y] (n-even) are very close to Bragg peaks, so in the case of certain tissue surfaces, it is preferable to replace the laser irradiation with particles such as electrons, protons, or ions.

Figure 2a, Figure 3a and Figure 4a present the graphs of H[1,y], H[2,y], and H[4,y] respectively. Numerous simulations of such functions show that for n-odd, all the graphs are well described by the functions close to zero along the *y*-axis, such as H[1,y]. On the other hand, the functions such as H[2,y] and H[4,y] are constants, and this is true for all the n-even. The above observations are made within the limits of computer performances and MATHEMATICA software 11 packages.

These observations are very important because it is only along the *x*-axis that the laser beam produces significant variations in the thermal fields. On the *z*-axis the situation is the same as on the *y*-axis, since the z variation is close to zero and therefore the temperature variation (absorption coefficient) is also very small (close to that of water).

Figure 3 shows analytical simulations in which all the parameters used in Figure 2 are unchanged, except for the irradiation time, which is 5 to 10 times longer than that used in the simulation in Figure 2. However, the comparison of the 3D and 2D plots from borg Figure 2 and Figure 3 shows that the temperatures behave non-linearly as the irradiation time is varied due to the quantum effect of the modified heat equation described in Equation (9).

The simulations performed with different laser powers show a direct proportionality between the incident laser power and the temperature variation of the target DNA molecule, with the same plot shapes and corresponding proportional values (Figure 4). This behavior is induced by Equation (7), which represents the solution of the heat equation. However, this model also has limitations induced by the heating laser intensity, which at values higher than 1 W leads to DNA degradation, making the model inapplicable.

The simulations shown in Figure 2a, Figure 5a and Figure 6a, the Hermite plots, designate the theoretical input data for the present simulation based on the laser intensity. The main graphs, Figure 2b, Figure 3a,c and Figure 4a,c, as well as Figure 5b and Figure 6b, present the 3D distributions of the thermal field as a function of space and time. It can be seen that for n-odd (Figure 2b), we have a temperature variation that is very suitable to use in DNA elongation without destroying the information contained in the DNA, which is fitting for its analysis. The thermal fields in the case of n-even are very high as variations for graphs close to x = 0, which is exactly the opposite situation compared to n-odd. On the other hand, if the laser intensity is increased slightly, the DNA information can be destroyed without causing very much damage to the living cells around the diseased cell. Future research in this field may take advantage of the fact that, in this situation, the thermal field is close to that of the Bragg peaks produced by irradiation with electrons, protons, and ions. The shapes of the graphs in Figure 2c, Figure 3b,d, Figure 4b,d, Figure 5c and Figure 6c depict the 2D density plot at the same temperature distribution. Figure 2d, Figure 5d and Figure 6d are variants of a 2D and 1D figure, showing the thermal field at a given time. Again, these graphics are limited by computer performance and software capabilities. It is noted that the 1D, 2D, and 3D thermal field distributions are in good agreement with experimental data from the literature [23].

Since the analytical simulations using the Hermite (3, y) plot have the same shape as the simulation in Figure 2, these were not presented in the paper.

## 5. Conclusions

A new generalized model based on the quantum heat transport equation is proposed to describe the interaction of ultra-short (fs) asymmetric moderate intensity laser beams with anisotropic DNA, in order to come closer to the actual experimental research. A highly accurate formula was derived to describe the temperature field in DNA samples during laser irradiation. The thermal distribution generated in the irradiated DNA strands was introduced by splitting the laser beam into transverse Hermite-Gauss modes. Fluctuations in laser intensity were accounted for by modeling them as superpositions of Hermite-Gauss laser modes.

It was demonstrated that only source terms of the form $a_{m}x^{m}$ do not exhibit circular symmetry, which proves to be an efficient computational model for molecular electronics simulations.

The new model can be rather easily extended to other ultra-short laser pulses or to interactions with substances of biological interest. This makes it easy to predict the important temperature variation due to fluctuations in the power of the incident laser beams. It should be noted that the previous version of these analyses [40] proved to be incapable of calculating the temperature field variation due to laser intensity fluctuations present in real experiments [43].

Precise values for the temperature field are required because DNA testing must be carried out without destroying the target information, and cancer treatment must locally destroy the affected DNA strands. It is important to note that, according to this study, the temperature variations do not exceed 10 K so that the DNA information can be preserved.

The simulations carried out with different laser powers show a direct proportionality between the laser power and the temperature variation of the target DNA molecule, presenting the same plot shapes and corresponding proportional values. However, as the irradiation time varies, the temperatures behave in a non-linear manner.

However, the model is also limited by the intensity of the laser beam because, at high intensities, the DNA breaks down and the model becomes inapplicable. In conclusion, the present paper deals with molecular electronics, presenting a new model that mixes the Kozlowski physical model with Zhukovsky’s recent radical approach to solving equations with partial derivates. This model is simplified and does not take into account the DNA’s internal structure. The proposed model has a fast simulation time of less than 1 min, which can be easily achieved on inexpensive commercial PCs. This avoids the need for complex and time-consuming software simulations such as molecular dynamics simulations [20].

We also suggest a replacement of electron, proton, and ion beam sources used in cancer treatment with laser beams.

## Figures and Tables

**Figure 1 materials-16-03334-f001:**
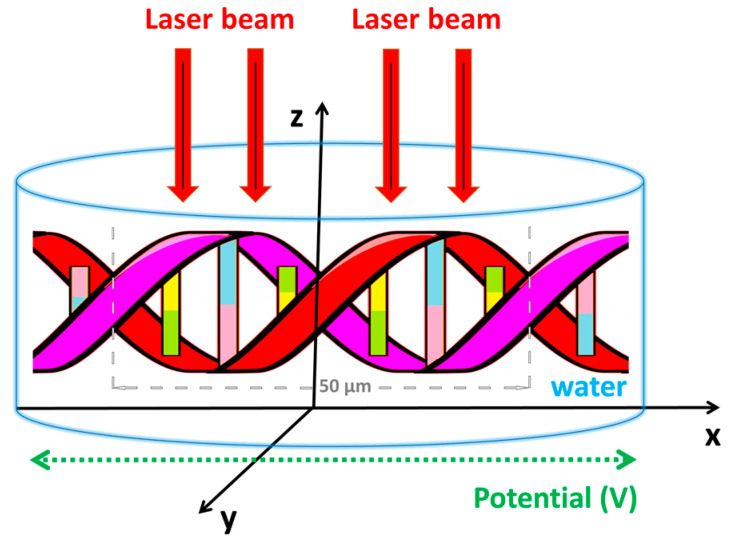
Schematic representation of the interaction between a laser beam and DNA.

**Figure 2 materials-16-03334-f002:**
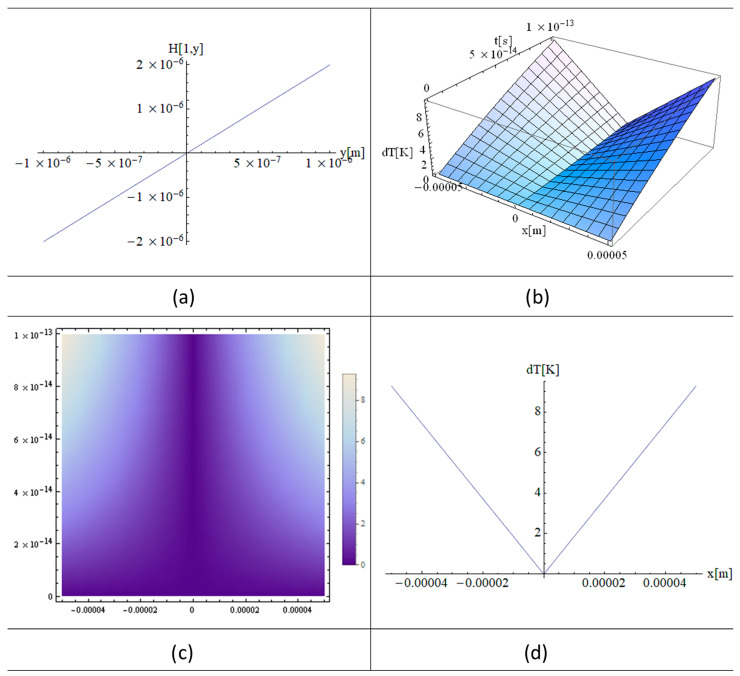
Analytical simulations using “MATHEMATICA” software 11: (**a**) Hermite (1, y) plot; (**b**) 3D plot; (**c**) 2D plot; and (**d**) 1D plot; using the following input parameters: T fields (**a**,**b**) in DNA with a power of *p* = 1 W and a laser irradiation time of t = 100 fs. Here, y and z ≈ 0.

**Figure 3 materials-16-03334-f003:**
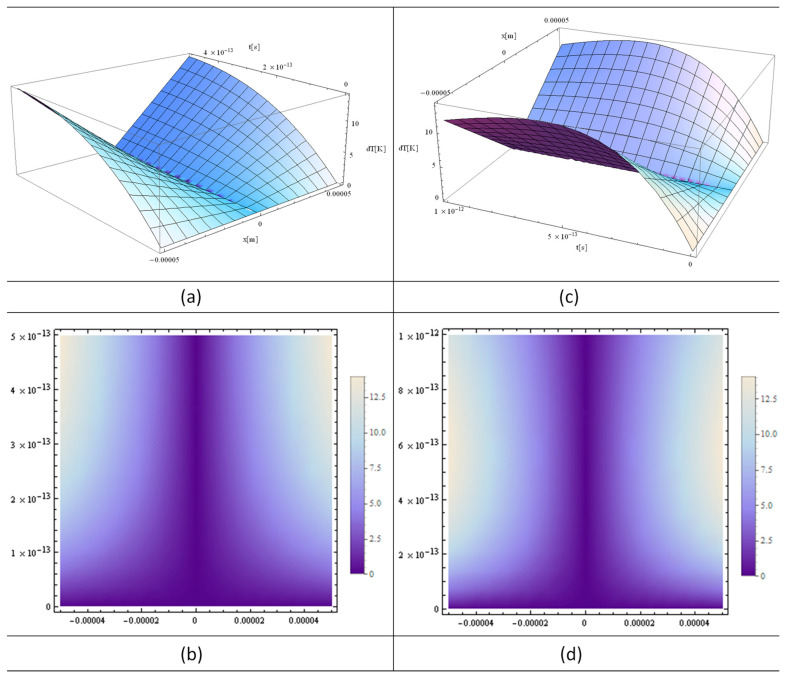
Analytical simulations using “MATHEMATICA” software 11: (**a**) 3D plot and (**b**) 2D plot for a laser irradiation time of t = 500 fs; (**c**) 3D plot and (**d**) 2D plot for a laser irradiation time of t = 1000 fs. The following input parameters were applied: T fields (**a**,**b**) in DNA with a power of *p* = 1 W; y and z ≈ 0.

**Figure 4 materials-16-03334-f004:**
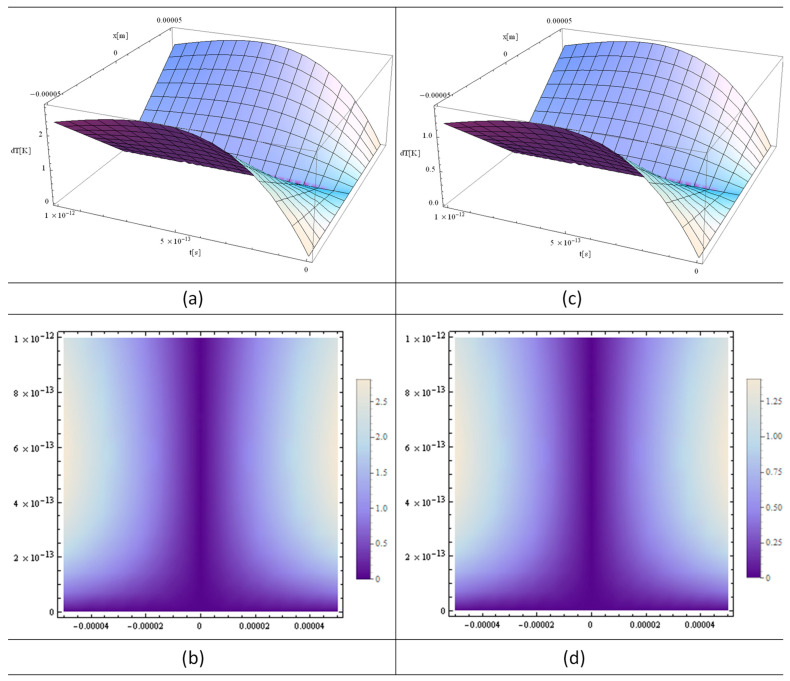
Analytical simulations using “MATHEMATICA” software 11: (**a**) 3D plot and (**b**) 2D plot for a laser incident power of 0.2 W; (**c**) 3D plot and (**d**) 2D plot for a laser incident power of 0.1 W. The following input parameters were applied: T fields (**a**,**b**) in DNA with a laser irradiation time of t = 100 fs; y and z ≈ 0.

**Figure 5 materials-16-03334-f005:**
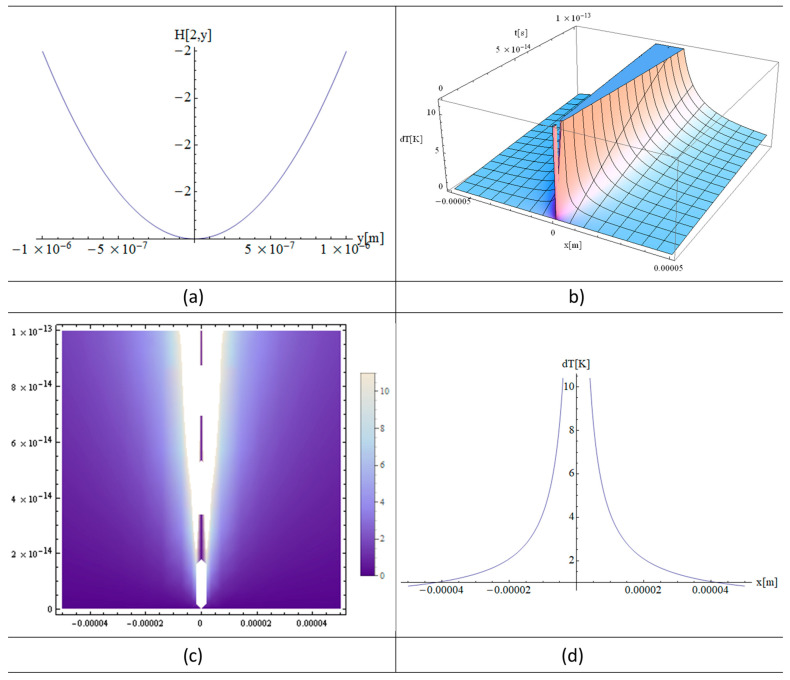
Analytical simulations using “MATHEMATICA” software 11: (**a**) Hermite (2, y) plot; (**b**) 3D plot; (**c**) 2D plot; and (**d**) 1D plot, using the following input parameters: T-fields (**a**–**d**) in DNA with a power of *p* = 1 W and a laser irradiation time of t = 100 fs. Here, y and z ≈ 0.

**Figure 6 materials-16-03334-f006:**
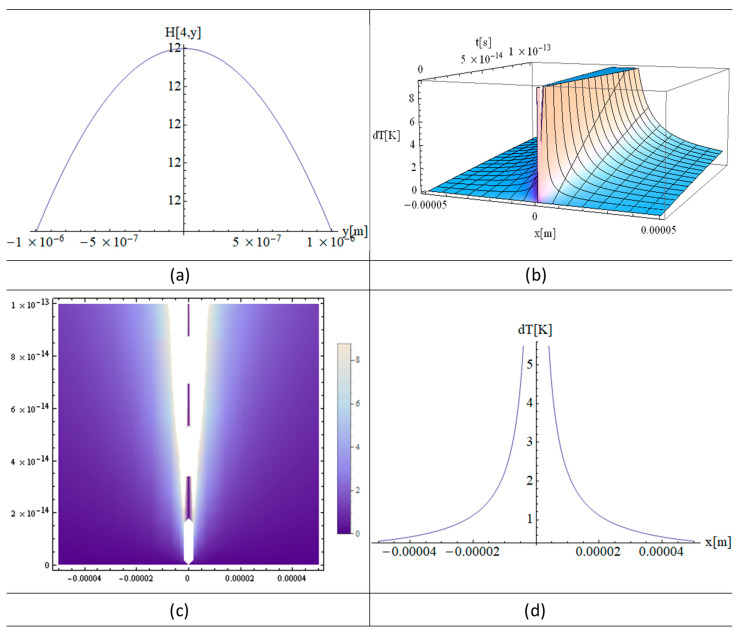
Analytical simulations using “MATHEMATICA” software 11: (**a**) Hermite (4, y) plot; (**b**) 3D plot; (**c**) 2D plot; and (**d**) 1D plot, using the following input parameters: T fields (**a**,**b**) in DNA with a power of *p* = 1 W and a laser irradiation time of t = 100 fs. Here, y and z ≈ 0.

**Table 1 materials-16-03334-t001:** Input data used in the MATHEMATICA software 11 [39].

Item	Value	Unit
Thermal wave velocity (υ)	0.05	nm fs^−1^
Characteristic temperature at the molecular level	316	K
Pulse duration	100	fs
Potential	3.25 × 10^−3^	eV

## Data Availability

The data presented in this study are available on request from the corresponding authors.

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
