# Peer review of "On the Thermal Behavior during Spatial Anisotropic Femtoseconds Laser-DNA Interaction: The Crucial Role of Hermite Polynomials"

_materials, 2023, doi:10.3390/ma16093334_

Round 1
Reviewer 1 Report
Suggestions or approaches obtained from simulation to be used in a real experiment setting need to be added to clearly explain the usability of this method. Please also add the limitations of this study regarding this.
What will happen if the environment of DNA under simulation is replaced with other than water? How robust is the simulation?
The author can add a table to compare similar simulations related to laser-DNA interaction and add the advantages and important features that can be obtained using the method suggested by the simulation.
Author Response
Thank you for taking the time to review our manuscript and for your helpful comments. We tried our best to respond to every comment or suggestion and modified the manuscript accordingly.
Q1. Suggestions or approaches obtained from simulation to be used in a real experiment setting need to be added to clearly explain the usability of this method. Please also add the limitations of this study regarding this.
A1: The introduced mathematical model is new and, according the best of our knowledge, similar models have not been available from literature. The closest model we could identify in the literature is given in Reference 38, cited in the text.
This note is inserted on page 3, paragraph 4, lines 6-8, where it reads as: “This contribution therefore aims to complement the existing knowledge with a completely new, powerful mathematical model to elucidate and explain the thermal behavior under irradiation of the DNA molecule.”
The main limitation of the presented model is related to the DNA collapse under intense laser irradiations, when the model becomes inapplicable.
This constraint was mentioned in the text on page 9, at the end of paragraph 2, where it reads as: “However, this model also provides limitations induced by the heating laser intensity, which at values higher than 1W leads to DNA degradation, making the model inapplicable.”
Q2. What will happen if the environment of DNA under simulation is replaced with other than water? How robust is the simulation?
A2: Water was chosen as the natural interaction medium because more than 73% of the human body tissue consists of water (consider Reference 18). The model simulation therefore assumes that the thermal parameters of the tissue are identical to those of the water.
This explanation is provided at page 6, Section 3 (Simulations), paragraph 2, lines 3-5, where it reads as: “… due to the fact that more than 73% of the human soft man body tissue is composed of water [40]. The model simulation therefore assumes that the thermal parameters of the tissue are identical to these of water.”
Q3. The author can add a table to compare similar simulations related to laser-DNA interaction and add the advantages and important features that can be obtained using the method suggested by the simulation.
A3: As mentioned in answer A1, the model presented is entirely new, therefore, so unfortunately it is not possible to include a table of similar simulations to illustrate possible advantages and other important features of our approach.
Reviewer 2 Report
The authors present a mathematical solution to approximate the heat transfer within a dsDNA irradiated by a pulsed laser source. My expertise is in the experimental side of the equation and as such I see the value in the area of investigation, though I cannot fully vet the mathematics described in the manuscript. Overall I think the ideas are useful but there are some aspects that need to be improved before publication.
11) The authors claim that the approach is useful and does not seem to be excessively expensive on the computation side, yet they only present a single example in their data. Where are the other laser powers? Different length pulses? I am also not clear what “DNA” is being modeled. Does it make a difference if it is CG or AT rich? I believe the authors should expand on this before the manuscript is accepted.
22) In section 2, when dealing with the x,y,z ranges, perhaps I am misunderstanding something but the diameter of dsDNA is generally around 2 nm, so why y,z values in the μm range? Similarly, dealing with equations 2 and 3, if DNA is basically rotationally symmetrical, why is there a difference between f(y) and f(z)?
33) The whole aspect of presenting this as useful for photothermal therapy seems unrealistic. How would you ever target DNA specifically with a laser when there is lots of other tissue that would absorb? I would suggest that the authors reference the idea of photothermal oligonucleotide release if they wish to focus on an application. I would particularly recommend the work of Hastman et al who have published on fs laser pulsed plasmonic-DNA systems (See below)
a. https://pubs.acs.org/doi/full/10.1021/acsami.1c19411
b. https://pubs.acs.org/doi/full/10.1021/acsnano.0c02899
Author Response
Authors are grateful to this Reviewer for her/his new comprehensive analyses of our previous text and the critical but constructive comments, suggestions and improvements, which we tried the best to consider in the revision of the manuscript. We answered point by point to the main points in her/his report while track changes have been used to mark our modifications in the main text.
Q1. The authors claim that the approach is useful and does not seem to be excessively expensive on the computation side, yet they only present a single example in their data. Where are the other laser powers? Different length pulses?
A1. As suggested, we conducted two new simulations series. First, we conducted two new simulations with different laser incident powers, for which we have presented the corresponding plots and interpretation in Figure 3 on page 9. These plots show a non-linear behavior with respect to the time of irradiation.
This explanation is provided at page 9, paragraph 1, where it reads as: “Fig. 3 shows analytical simulations in which all the parameters used in Fig. 2 are unchanged, except for the irradiation time, which is 5 and 10 times longer than that used in the simulation in Figure 2. However, the comparison of the 3D and 2D plots from borg Fig. 2 and Fig. 3 shows that the temperatures behave non-linearly as the irradiation time is varied due to the quantum effect of the modified heat equation described in equation (9).”
We then ran another series which showed a direct proportionality between the laser power and the temperature variation of the DNA target. The results are shown in Figure 4 on page 10 of the revised manuscript.
The explanation of the figure is provided in text at page 9, paragraph 2, where it reads as: “The simulations performed with different laser powers show a direct proportionality between the incident laser power and the temperature variation of the target DNA molecule, with the same plot shapes and corresponding proportional values (Figure 4). This behavior is induced by equation (7), which represents the solution of the heat equation.”
Q2. I am also not clear what “DNA” is being modeled. Does it make a difference if it is CG or AT rich? I believe the authors should expand on this before the manuscript is accepted.
A2. We simulate a DNA molecule with a high number of DNA double strands immersed in water. Since more than 73% of the soft human body tissue consists of water, the water was chosen as natural interaction medium. The model therefore assumes that the thermal parameters of the tissue are identical to those of the water.
This explanation is provided at page 6, Section 3 (Simulations) paragraph 2, line 3-5, where it reads as: “… due to the fact that more than 73% of the human soft man body tissue is composed of water [40]. The model simulation therefore assumes that the thermal parameters of the tissue are identical to these of water.”
The model doesn’t distinguish between very small sequences such as CG or AT, as no significant modifications are expected when the results are compared with CG or AT enriched ADN samples.
This note is inserted on page 3, at the end of paragraph 4, where it reads as: “The model is simplified and doesn’t take into account the internal structure of DNA, as no significant modifications are expected when the results are compared with nucleotide sequences with enriched cytosine-guanine or adenine-thymine partitions.”
Q3) In section 2, when dealing with the x,y,z ranges, perhaps I am misunderstanding something but the diameter of ds DNA is generally around 2 nm, so why y,z values in the μm range? Similarly, dealing with equations 2 and 3, if DNA is basically rotationally symmetrical, why is there a difference between f(y) and f(z)?
A3. According to Masatoshi Ichikawa et. all (Reference 21), the diameter of a DNA molecule can expand over a quite large range from a few nm to tens of μm, even exceeding 50 μm. This is due the assumption that it contains several μm-sized random coil conformations in solution due to entropic elasticity.
This note was inserted on page 4, paragraph 4, where it reads as: “According to the current literature, the diameter of DNA molecules covers a rather large range, from several nm to tens of μm, as it is thought to contain several random coil conformations in solution due to entropic elasticity, and thus containing a number of more than 7000 DNA double strands [Reference 23]. The model therefore assumes that the length of the DNA molecule is 50 μm.”
Next one should emphasize that although DNA molecules are symmetrical along the y and z axes, the system is however asymmetric because the direction of the laser beam propagation is along the z-axis, but not along the x or y axes.
This explanation was introduced in the text on page 3, Section 2 (Mathematical Modeling). Paragraph 1, lines 2-4, where it reads as: “Although DNA molecules are symmetrical along the y and z axes, the system behaves asymmetrically because the direction of the laser beam propagation is along the z-axis.”
Q4) The whole aspect of presenting this as useful for photothermal therapy seems unrealistic. How would you ever target DNA specifically with a laser when there is lots of other tissue that would absorb? I would suggest that the authors reference the idea of photothermal oligonucleotide release if they wish to focus on an application. I would particularly recommend the work of Hastman et al who have published on fs laser pulsed plasmonic-DNA systems (See below)
- https://pubs.acs.org/doi/full/10.1021/acsami.1c19411
- https://pubs.acs.org/doi/full/10.1021/acsnano.0c02899
A4. Our model is simple, and doesn’t take into account the DNA internal structure, but the DNA as a molecule. We also intend to target localized cancer cells that show no signs of having spread. Therefore, the simulation was performed for DNA molecules as a bulk, in water, so that even if some of the radiation is absorbed by the tissue, the final result is not significantly affected.
Also, the proposed model gives a fast simulation time, less than 1 min, which can be readily obtained in inexpensive commercial PCs. Thus we avoid complex and time consuming software simulations such as molecular dynamics simulations.
We have also added a few new references in the text, including one of the ones mentioned above.
This explanation was introduced in the text on page 3, paragraph 3, where it reads as: “The model describes the interaction of ultra-short (fs) laser beams of asymmetric moderate intensity with the DNA molecules to analyze the thermal changes and proposes a model to eradicate the diseased localized cancer cells that show no signs of having spread.”
The explanation continues at page 3, at the end of paragraph 3 as: “The model is simplified and doesn’t take into account the internal structure of DNA, as no significant modifications are expected when the results are compared with nucleotide sequences with enriched cytosine-guanine or adenine-thymine partitions. Also, the model gives a fast simulation time of less than 1 minute, and can be easily adapted for more complex biological simulations.”

Round 2
Reviewer 1 Report
The revision has been provided